# Facilitators, Barriers and Views on the Role of Public Health Institutes in Promoting and Using Health Impact Assessment—An International Virtual Scoping Survey and Expert Interviews

**DOI:** 10.3390/ijerph192013367

**Published:** 2022-10-16

**Authors:** Liz Green, Kathryn Ashton, Lee Parry-Williams, Mariana Dyakova, Timo Clemens, Mark A. Bellis

**Affiliations:** 1Policy and International Health, WHO Collaborating Centre on ‘Investment in Health and Well-Being’, Public Health Wales, Cardiff CF10 4BZ, UK; 2Department of International Health, Care and Public Health Research Institute CAPHRI, Maastricht University, Duboisdomein 30, 6229 GT Maastricht, The Netherlands

**Keywords:** health impact assessment, public health institutes, health in all policies

## Abstract

Public health institutes have an important role in promoting and protecting the health and well-being of populations. A key focus of such institutes are the wider determinants of health, embracing the need to advocate for ‘Health in All Policies’ (HiAP). A valuable tool to support this is the health impact assessment. This study aims to support public health institutes to advocate more successfully for the use of health impact assessments and HiAP in order to promote and protect health, well-being and equity. During July 2021, a quantitative online survey was undertaken across international networks with 17 valid responses received. Semi-structured interviews were also administered with nine expert representatives and analysed thematically. In total, 64.7% (*n* = 11) of survey respondents were aware of health impact assessments and 47.1% (*n* = 8) currently conducted health impact assessments. It was noted that there are differing approaches to HIAs, with a need for a clear set of standards. Barriers to use included lack of knowledge, training and resources. Overall, 64.7% (*n* = 11) of survey respondents would like to do more to develop knowledge and capacity around health impact assessments. The results from this study can serve as a platform to help build knowledge, networks and expertise, to help support a ‘Health in All Policies’ approach and address inequalities which exist in all societies.

## 1. Introduction

Public health institutes (PHIs) have an important role to promote and protect the health and well-being of populations at a national (or sometimes regional) level [1]. They have a remit to monitor health, establish and gather evidence, provide advice and protect the population from communicable and non-communicable diseases [2]. They also have a role in providing guidance on health improvement and promotion such as smoking cessation or workplace and school health-based programmes and have workforces with essential skills from a range of health disciplines which can assist in setting national policy direction [1]. PHIs can be defined as ‘an organizational unit of a national government health ministry (not of a state or province), which serves the whole country as a source of technical public health expertise and would be the unit called upon to respond to public health threats’ [3], whilst the International Association of National Public Health Institutes (IANPHI) defines a PHI as ‘a government agency, or closely networked group of agencies, that provides science-based leadership, expertise, and coordination for a country’s public health activities’ [4]. Advantages of a PHI include the assembly of a stable mass of expertise, continuity of experience, and the scientific knowledge and appropriate human, technical and financial resources to tackle public health challenges; it is an independent scientific organization without political affinity; increasingly, PHIs have a role to work on behalf of their country on public health issues that cross national boundaries [3].

During the 20th century, PHIs were established to support and address critical public and environmental health-related issues, such as infectious disease outbreaks and sanitation conditions which could affect health [3,5]. The majority of PHIs’ work remains focused on communicable disease control and environmental health protection, and this was reinforced during the COVID-19 pandemic. The recent pandemic has also highlighted the critical importance of PHIs and the key expertise and skill sets held within them to address important health threats [2]. However, presently, the range of public health activities within a PHI can vary and include a focus on advocating for health and well-being at a national level, reducing health inequalities, tackling non-communicable diseases such as obesity, and policy work [2,6]. They also include the same historical focus on health protection and communicable diseases (including immunizations, epidemiology and infectious diseases), environmental health and safety, and health services research [3]. Some PHIs have broadened their work approaches to include the wider determinants of health and how these will affect population health and inequalities [2,7]. In doing so, they have embraced the need to advocate for a consideration of ‘Health in All Policies’ (HiAP) [8,9,10] in order to address the causes of poor health—traditionally described as the ‘causes of the causes’ [11,12]. They have taken specific perspectives on how to tackle these—by engaging with decision makers, providing evidence and health intelligence data and also through the use of specific tools to help assist policy and decision makers to better understand the implications of their decisions [13,14].

A key tool and vehicle to support, drive and implement HiAP is the health impact assessment (HIA) [15]. HIAs are a widely used methodology, commonly defined as ‘a combination of procedures, methods, and tools by which a policy, intervention or service may be judged as to its potential effects on the health of a population, and the distribution of those effects within a population’ [16]. As a flexible tool which can be applied proportionally in practice, an HIA allows health and well-being to be considered in all policy areas such as planning or housing as a method of implementing a ‘Health in All Policies’ approach and has the power to influence the decision-making process by promoting cross-sector collaboration [17].

There is little peer-reviewed published academic research which illustrates the use, impact and co-benefits of HIAs which can be reaped by PHIs promoting and using HIAs beyond a few recent case studies [18,19,20]. There have been recent examples of surveys which provide a snapshot of global HIA practice, but they did not specifically focus on how the methodology can be promoted or used by PHIs [21,22]. Whilst some PHIs promote the use of HIAs as a way to identify the wider health, wellbeing and equity impact of policies, plans and projects on the population [23,24,25], there is a need to further explore and understand how PHIs globally are currently using and promoting HIAs, if at all, what the barriers and enablers are, and what can be done to promote and use HIAs more in the future. The COVID-19 pandemic has also presented an opportunity to review work streams within PHIs as they, and society, advance into the pandemic recovery stage [2,4].

This study aims to support PHIs in their capacity and capability and strengthen their ability to advocate more successfully for the use of HIAs and HiAP in order to promote and protect health, well-being and equity. This paper outlines the results from an online survey and interviews undertaken with representatives from PHIs to inform future practice in HIAs for PHIs, and to share learning from each other. It develops a base for a shared understanding, paints an international picture of HIA practice, and can lead to future work at a global and national level.

## 2. Materials and Methods

A digital international survey and interviews were carried out to scope and capture global HIA practices in PHIs, how they are being implemented (if at all), and any challenges, enablers and opportunities. The interviews provided an opportunity to explore in more depth issues raised in the survey and understand the nuances of practice and the different positions and priorities for PHIs in respect to HIAs. The survey was targeted towards, and disseminated to, national and regional PHIs. When defining the sampling frame, the following definition of PHIs was utilized:

‘A Public Health Institute (PHI) is a government agency, or closely networked group of agencies, that provides science-based leadership, expertise, and coordination for a country’s or region’s public health activities’ [3,4].

Third sector bodies, and other organisations with a public health focus were deemed to be outside of the scope of this research. In addition, this study was part of a wider joint project which aimed to scope understanding and use of social value methods within PHIs. Results from that study have been published elsewhere [26].

### 2.1. Survey Design and Dissemination

A self-administered quantitative virtual survey was disseminated using Survey Monkey during July 2021. The questionnaire asked about background details for the respondent’s PHI, their HIA awareness and experience in their PHI, and any barriers and facilitators to using the process. Of the total maximum of 45 questions included in the survey (some were only asked if respondent answered yes), 10 questions were open-ended, and 35 were of a closed format (see Appendix A for the questionnaire). Due to resource limitations, the survey was only made available in English. The survey was internally tested within a PHI and also externally with IANPHI. Feedback was considered and incorporated into the final survey.

For the dissemination of the survey, two non-probability sampling methods, namely purposive and convenience sampling, were used [27]. Responses were only included if the respondent was an official from a PHI with a national or regional portfolio. An invitation and a participant information sheet were circulated by email via a range of networks. These included IANPHI, World Health Organization (WHO) networks including the Regions for Health Network and EuroHealthNet. The questionnaire was also directly circulated to previously identified representatives from PHIs (convenience sampling). At the midpoint of the data collection period, reminders to gather more responses were sent.

As indicated in the NHS Health Research Association ethics decision tool [28], approval from an NHS Ethics Committee was not needed for this study to be undertaken. It posed little potential harm to those taking part, and all the data which was collected and analysed was anonymised and safely secured digitally to protect personal data and privacy.

### 2.2. Semi-Structured Interviews

At the end of survey completion, all respondents were asked whether they would want to take part in a semi-structured interview to further the conversation. Questions in the interview guide were steered by the survey results and aimed to allow for triangulation of results. The semi-structured approach allowed participants to demonstrate their views and experience, but also helped gently guide specific areas of interest for the researcher. Individuals who agreed to participate were invited to participate via email, and informed consent was obtained prior to interviewing. Interviews were conducted via virtual video calls. The interviews were recorded digitally, and a professional transcription company transcribed and anonymized them.

For the survey closed question responses, analysis was carried out using Microsoft Excel. The responses from the open-ended questions in the survey and data from the interviews were analysed thematically by two researchers.

## 3. Results

### 3.1. Study Participants

The survey was live from 7–19 July 2021 and gathered 37 responses. Unfortunately, 12 (29.7%) needed to be excluded from analysis due to either being incomplete responses or because the organization they were representing did not fit the inclusion criteria. Within the remaining 25 responses that were eligible to be included, 17 countries were represented. In a few cases, countries or regions had more than one response. To minimize the introduction of country bias, responses from the same country were amalgamated. In total, 76.5% (*n* = 13) of respondents were based in PHIs in Europe (Norway, the Netherlands, Portugal, Ireland, Italy, Scotland, Spain, Belgium, Republic of Moldova, Finland, Iceland, Wales and Sweden), 11.8% (*n* = 2) were based in Asia (South Korea and Israel) and 11.8% (*n* = 2) were based in Oceania institutes (New Zealand and Australia). There was an absence of responses from North America, the Middle East and Africa despite attempts to make contact via email.

There are 110 NPHIs registered as members of IANPHI. The 17 country responses to this scoping survey provided a 15.4% response rate of all members of IANPHI. Given the exploratory, first-step nature of this research, this is a reasonable response rate. Of the survey respondents, 11 stated within the survey that they would like to take part in an interview. After further communication, during September and October 2021, a total of nine interviews were carried out. This was deemed to be a satisfactory number of interviews due to the scoping and exploratory nature of this work, and saturation point was reached with no new information being provided due to the specificness of the topic. The interviewees represented nations such as Australia, Portugal, Iceland and the UK and Ireland including devolved nations such as Wales and Scotland. The roles of those interviewed included chief executive, public health specialist, programme manager, environmental health specialist and HIA specialist.

### 3.2. Raising Awareness and Promoting an Understanding of the Key Concepts of HIAs

Amongst survey respondents, the level of awareness of HIA methodology prior to being asked to complete the survey was 64.7% (*n* = 11). A total of four interviewees stated that there was a high awareness of HIAs within their organisations. Interviewees indicated that they promote HIAs (when they can) in a range of ways, for example, using existing evidence resources and public health indicators to promote the importance of HIAs and aligning work with the Sustainable Development Goals and other key policies. Several participants highlighted the work they are doing in this arena, which includes trying to develop more guidance and tools and use existing case studies (even if they are derived from other nations) to promote HIAs and their benefits.

### 3.3. Why Are PHIs Important in the Use and Promotion of HIAs?

Five of the nine interviewees believed that PHIs are important in the use and promotion of HIAs because their PHI has an environmental determinant and health protection focus and an ability to statutorily respond to environmental impact assessments (EIAs). In addition, four interviewees stated that PHIs were important due to the fact that they have a clear remit for wider public health, prevention and HiAP approaches and inequalities. This gave them legitimacy and credibility when promoting HIAs as a tool to inform decisions in advance of making decisions.

I think because they’re normally government led, they have a bit of, they have the, again, it’s that credibility and the endorsement and the recognition amongst other government agencies. (Interviewee I)

Two interview respondents stated that the latter then enabled them to start conversations around health and another two that PHIs hold the competence, core knowledge and skill sets and evidence to support the use and promotion of HIAs. Interestingly, only two stated that their PHI could lead in their context by setting out clear direction about HIAs and advocating a consistent approach and methodology.

Otherwise, it’s just another metric that, you know, lots of consultancies and people, you know, have vary, variations that you could describe as HIA, or different frameworks that just, you know, decision making frameworks that they’ve made up, you know, they’ve kind of developed themselves. There’s nothing wrong with that, but we want everybody to be doing the same thing. (Interviewee I)

### 3.4. Current Use of HIAs in National and Regional PHIs

With regards to the use of HIAs as a method of assessing the impact on health, interviewees from four countries indicated in the survey that HIAs are mandatory in the environment field of application, for example, environmental impact assessment or strategic environmental assessment. This was at both a national and regional level in three of the countries, with one reporting it was mandatory at a national level. In addition, just under half (41.2%) of the institutes who responded to the survey reported having a lead for ‘Health in All Policies’. In total, 35.3% (*n* = 6) had a dedicated in-house lead/resource for HIAs. Of those, four (23.5%) had an HIA toolkit or guide, three (17.6%) stated their HIAs had a quality assurance process, and nine (52.9%) currently advocated for HIAs.

In total, 47.1% (*n* = 8) stated in the survey that their institutions currently undertook HIAs, but the subject matter and level differed. This ranged from air quality, transport, housing and spatial planning through to health service interventions and mental well-being initiatives and national policies such as climate change and COVID-19 pandemic measures. Eight survey respondents (47.1%) stated that HIAs were primarily used to support EIAs or concentrated on environmental health determinants for example, air quality, whilst only four (23.5%) specified that they focused on wider determinants of health and mental wellbeing and wider policies and services, for example, the impact of ‘lockdown’ on the population. These data were reinforced in the interviews. Seven interviewees stated that when HIAs were carried out, they were very ad hoc, across differing teams, or EIA-focused. Only one PHI reported having a dedicated HIA team and director for HIAs.

This differs across the organisation…. We are more likely to undertake Health Inequalities Impact Assessments on our own programmes of work, and to support HIAs undertaken by others. But ….has done a range of HIAs on things like housing and planning. Our environmental health team have done them on air quality and I think maybe alternative heat sources. (Interviewee C)

### 3.5. Perceived Barriers to the Use and Promotion of HIAs

In the survey, reported barriers to the use of HIAs or their promotion included that they are not an area of prioritisation at present (*n* = 7, 41.1%), lack of knowledge (*n* = 4, 5.88%), lack of training (*n* = 6, 35.2%), and lack of ability to advocate for them (*n* = 2, 11.7%). By far the most cited reason (70.5%) was a lack of resources (*n* = 12), which was reiterated by the interviewees. The need for more HIA training was emphasised by six of the nine interviewees:

You know, a lot of, additional learning through conferences, seminars, e-learning, the different kind of routes that people have for the existing workforce, and then I suppose for the workforce coming through, it’s about embedding that more strongly into Public Health Training and things like Masters in Public Health and epidemiology. (Interviewee C)

There is great interest in this work, we have strong group, but it would help to have further training and more time available to do the work. Other tasks are not going away. (Interviewee D)

Three interviewees also highlighted the lack of, and need for, political leadership or stewardship in this space and how it could make a huge difference in terms of enabling and creating a positive environment for HIAs or hindering their implementation. Two-thirds of interviewees cited financial resources as a barrier to building capacity to advocate for and develop HIAs.

Despite theoretical institutional interest on HIA, the lack of resources, leadership and clear political support are just some of the very difficult barriers lived by organizations to undertake a real institutionalization of HIA. (Interviewee E)

It shouldn’t be the most important. But at the end of the day, it’s it is important, isn’t it, but where, where is the money coming from? (Interviewee G)

Finally, it was highlighted by two interviewees that it is perceived as ‘another process to do’ and, therefore, not be meaningful by becoming a ‘tick box exercise’.

…there is always the problem, I suppose not a problem, well I don’t see it as a problem, but what some people would see is HIA, the only pushback we’ve ever received on it is the fact that, oh my…, it’s just another process and thing to do. (Interviewee I)

Legislation could support progress around this, but the type of legislation and nature of it was also cited by three interviewees as a barrier. Lack of legislation led to HIAs not being carried out routinely across the PHI but also the locality.

Then we don’t have legal support, a legal-law, a law that says that you need to do this. But I think in sometimes it’s not only the law, it’s the perception of it, because we have this EU directive, which say that we need-there’s a, a description of health in the, in the impact-health impact-not health-environment impact assessment. (Interviewee E)

### 3.6. Perceived Benefits to PHIs

Interviewees expanded on the benefits of HIAs as a follow on from questions about the barriers. Six interviewees of the nine stated that HIAs are beneficial to PHIs because they focus on prevention as an ‘ex-ante’ tool.

HIA is seen very much as it’s become more popular as pre, you know, as it’s supposed to do, pre-empting problems that come down the line, and I think there’s an attempt to have a more holistic approach to dealing with health inequalities, in particular. (Interviewee I)

Five of the interviewees highlighted that HIAs could facilitate HiAP by considering health in other/traditionally described ‘non-health sectors’ such as spatial planning or housing, and four noted that HIAs can have a clear focus on wider determinants of health.

…HIAs… are very useful into bringing, so again, whether we call it health in all policies …or a cross sector sort of engagement is where of course, we try to influence the wider determinants of health, it is a very helpful because on one hand, they show the different sectors, what is their impact on health, but also they show with the, with the social value as to why, what is what Public Health is doing, which can actually has a value to their own areas of responsibility. (Interviewee G)

Five interviewees also identified that HIAs can improve plans, strategies and projects to make them healthier and two stated that they do this by building health into decision making.

Also, the thing about using, you know, like, if you use HIA as a tool to look at, like a built environment intervention or something like that, then, yeah, it provides a way for the Institutes to get that into part of the decision-making process, and, yeah, I think that’s probably one of the key aspects of it. (Interviewee A)

Five interviewees noted that HIAs address equity and inequalities by considering population groups as part of the process. Other reasons provided included that HIAs can be a useful involvement tool through which to engage with a variety of stakeholders and start conversations about health and inequalities.

The identified barriers and benefits of HIAs from the survey are demonstrated in Table 1.

### 3.7. What Could Be Done to Improve the Situation and the Awareness and Enable the Use of HIAs in NPHIs?

In total, 64.7% (*n* = 11) of survey respondents would like to do more to develop knowledge and capacity around HIAs in their institutes. The survey responses provided insight into how PHIs could improve awareness of HIAs, their effectiveness and their outputs and benefits.

#### 3.7.1. Improving Awareness of HIAs as a Methodology

Open-ended survey responses included five respondents (29.4%) citing the need to embed HIAs in public health training and education; four (23.5%) citing awareness-raising including case studies and conferences which showcase the role of HIAs in policy development; two (11.7%) stating government support, two (11.7%) stating having identified centres of excellence and one (5.9%) citing having dialogue with commissioners.

The participants in the nine interviews supported all of these and articulated them further.

And if you’ve got someone good from public health, who knows how to communicate the benefits to people, you can actually have really good conversations that help people in other sectors to understand why they have a role to play in health improvement and why they… What they can do, I suppose, so, you know, people don’t generally….don’t generally want to harm health. (Interviewee C)

So they, they always look for, kind of lots of evidence to deliver something, case studies. But the, the major thing I’ve always found is that if it’s something that’s been transferred from a similar jurisdiction and it has worked well there, that, that really gives them a lot of confidence in, in pursuing it. (Interviewee I)

I think the Welsh example of having legislative mandate and also a lead agency is an excellent example of how to strengthen HIA across all the dimensions. (Interviewee A)

#### 3.7.2. Improving Awareness of HIA Outputs

In terms of awareness-raising of HIA outputs, the open-ended survey responses revealed several themes. These included the belief that embedding HIAs in core business would lead to more officers being familiar with HIAs and their outputs, cited by four respondents (23.5%); the need for stronger regulation, cited by three respondents (17.6%); two (11.7%) stated the need for more high-quality evidence to support HIAs, two respondents (11.7%) believed in increasing the awareness of the role of NPHIs in HIAs and decision making, and successful case study examples were also cited by two (11.7%) as a method of improving HIAs and their outputs.

### 3.8. What Support Would Public Health Officers in Phis Need in Practice to Promote and Use HIAs?

The interviews reinforced the survey results described above with four interviewees stating that having buy-in from key stakeholders and politicians was important, including legislation for HIAs.

And at the top, yes, we’ve got buy-in, I would say, and this period (COVID-19) has really cemented that buy-in in terms to the concept because of the work that’s gone on and they’ve seen the value in it. (Interviewee B)

Four interviewees identified learning from, and highlighting, the work in Wales and Public Health Wales’s Wales HIA Support Unit and noted that following a similar model would help them.

It’s been very helpful the work that Wales has done on everything. We really- we know we have a place to look for… But we- And with this COVID-19, we have really been looking into the work … done for that and are… adapting, you know, to [inaudible-0:27:18.3] following the steps… (Interviewee D)

Case studies, always invaluable. I mean, you know, what we’ve learned from x’s work in Wales has been really, really helpful to us, you know, and that really is what you need. You know, you need somebody who’s gone down the path before you. We’re modelling completely on what Wales’ team has done. (Interviewee I)

Three interviewees stated that more resources would help along with three who believed that increasing capacity and allowing the time to learn or apply their skills to HIAs would increase their confidence in promoting and using the process. Other reasons included embedding HIAs in core business of the institute and highlighting the environmental, economic and social value that a process such as the HIA can support.

But… I don’t think we have to make the case anymore…. What we have to do is really, truly embed it into people’s practice. (Interviewee C)

### 3.9. COVID-19 Pandemic

Finally, the survey asked if there were any further comments, and this revealed some reflections derived from the COVID-19 pandemic. Both within the survey open-ended questions and the interviews, six survey participants identified how HIAs had captured the wider impact of the pandemic and that their way of capturing the health impacts was beneficial to broaden the conversation around health and equity, and three referred to this in the interviews.

‘it was difficult for it (HIA) to get traction. I think it’s beginning, I think the time is right now for it to get traction because people are beginning, post-pandemic, particularly to understand more about social determinants of health, about how, you know, personal responsibility is not the only issue’. (Interviewee I)

## 4. Discussion

This study captured some of the barriers and enablers for HIAs in PHIs and highlighted how HIAs are being promoted or used currently (or not) by them across the world. It highlighted how PHIs could further advance the work they do to increase an awareness of HIAs, promote better understanding of the tool and its use at a regional or national level and better understand the barriers which they may currently face and opportunities which can be utilized, for example, any requirements for a consideration of health in other assessments such as EIAs which may be legally required. Both the survey and interview respondents recognised HIAs as an important tool to drive HiAP approaches to improve health and reduce inequalities [8,15,29]. HIAs were also recognised as a prevention tool which identifies and anticipates any negative impacts which can be mitigated as part of the journey of policy or project development.

This scoping study highlighted how HIAs have been utilised in responding countries. This is consistent with previous surveys which have looked at international HIA practice [28,30]. The lack of engagement from PHIs outside of Europe, for example, North America and Africa, indicates this, although some work has been carried out in these geographical regions [31]. Although the work is exploratory and the numbers are small, what is new is the insight provided into how institutions such as PHIs view HIAs, the perspectives and approaches they do or do not take, and how they look for clear examples of successful practice to assist them. It can provide a direction of travel for future work with both PHIs and the wider HIA community.

A good overall awareness of the concept of the HIA was reported in the research, including how the approach and methods could be utilised in practice. However, it is important to note the differing approaches to HIAs described by PHIs who engaged in the survey and interviews, with some very focused on environmental determinants of health and health protection risk such as transport or air quality and using EIAs as a vehicle to address and mitigate health risk. This is compared to some other PHIs who emphasise a social determinant of health and equity approach to HIAs and health maximisation. Although this distinction in practice has been described previously [28,30,31,32], it did not focus on PHIs specifically and included practice from a range of public and private organisations such as private environmental and health consultancies. This is an important distinction to draw in the practice of HIAs, as many in the survey referred to the focus on environmental determinants of health and responding to EIAs. This could highlight a potential key issue, where many PHIs may think of HIAs as synonymous with EIAs. This could, therefore, be limiting where and how PHIs deploy the HIA process particularly in nations where EIAs are already established in legislation. It could also highlight the issue of methods of collecting evidence, confidence in them and the acceptability and accuracy of the evidence and data. For some PHIs responding to or supporting consultations and decision-making processes, EIAs and other IAs may be viewed as being more robust, valid and acceptable by containing more technical health data and epidemiological evidence, compared to more qualitative or mixed-method HIA processes [33,34,35]. Therefore, the perspective, context and remit of the PHI are very important to HIA practice and setting strategic direction for the process.

Barriers to current use of HIAs reported in this research included the lack of understanding of how HIAs can benefit a wide range of sectors’ work by improving plans or policies, and a lack of understanding in institutes about the benefits of the process which will assist them to carry out and advocate for HIAs [33]. This scoping study also reinforces the fact that lack of knowledge, training and resources constrains the use and promotion of HIAs in many PHIs even if they have received training, as they still lack confidence. This could be due to HIAs only being one part of their job role and being seen as ‘something else to do’ which risks becoming a ‘tick box exercise’. Many of the survey respondents and those interviewed referred to the need for additional training and capacity in order to carry out or promote HIAs; this has been previously cited as a barrier to the implementation and effectiveness of HIAs [28,36,37]. A previous study carried out specifically for public health officers and spatial planning identified that ‘training provision for Public Health Practitioners in reviewing spatial planning HIAs was found to be very limited, with 65% reporting that they had received no formal HIA training’ [7]. Therefore, despite the advancement and evolution of HIAs and tools, practitioners still identify the need for more training and resources. Institutional support and resources would be welcomed by PHIs, including support from organisations such as the World Health Organization in advocating for HIAs and learning from expert PHI teams such as that in Wales [24].

However, some of these barriers could be overcome with increased promotion inside and outside the PHI, as well as joined up working with likeminded practitioners and officers based in PHIs to carry out small-scale HIAs, which are not time or resource intensive. This would enable them to familiarize themselves with the process in practice, ‘learn by doing’ and gain confidence to carry out an HIA and importantly advocate for it as a process. This has been recognized as an important first step [20,21]. It would form a rich learning experience and provide a case study example to ‘sell’ to the organization (and politicians who can make it a legal process) and explain any benefits gained from it. This has been the experience in Wales [38]. At the same time, strategic advocates for HIAs need to be created within an organization, and engaging with senior leaders and demonstrating the value of standalone HIAs as part of prevention strategies and HiAP approaches to them is hugely important. This is because they have the power to commission and allocate resources for HIAs. A ‘bottom up’/‘top-down’ approach is necessary, as is utilizing all facilitators and enablers which officers may have already, for example, the inclusion or strengthening of a consideration of health in EIAs. In addition, collaboration and learning between, and from, PHIs and the sharing of virtual training sessions and resources which would be time effective and resource efficient is a way forward. HIAs as an impact assessment can also learn from the evolution and development of other impact assessment processes [39]. Practitioners can also learn from different perspectives that decision makers and legal processes apply about evidence in IAs, for example in EIAs, which include more health quantification data and epidemiological exposure and estimate studies which can increase the validity of the findings and provide more confidence in them [33,34].

Whilst several barriers were highlighted, many participating institutes felt positive about the use, and advocacy of, HIAs and the benefits they can bring, with several interviewees citing the distinct example of PHIs such as Public Health Wales’s work and the Wales HIA Support Unit, as a centre of practice to replicate. There was a consensus that carrying out HIAs can improve planning, policy delivery and design by prospectively anticipating issues and addressing them in an evidence-based way. It can promote a consideration of health and well-being in decision making. It was also noted that they involve wider sector stakeholders, for example, spatial planners to facilitate conversations about health inequalities and enhance engagement. It is a clear driver of the concept of HiAP in doing this—by seeking synergies with others and anticipating impact in order to reduce negative impacts and inequalities—and can reap benefits by making others understand their impact on health and equity status [29,31].

Legislation was clearly cited in the interviews as being an enabler—if in existence—but an inhibitor if not. EIA directives, Public and Environmental Health and other IA legislation have provided opportunities [28,30] for health inclusion and PHI input. Sharma et al. [7] reinforced the need for statutory levers, but without confidence to engage or lack of knowledge and resources, PHIs are hampered. In the Welsh context, legislation specifically making broad HIAs a statutory requirement for public bodies [40,41] and the Wales Health Impact Assessment Support Unit (WHIASU) and its model of being situated within a PHI (Public Health Wales) were cited several times in interviews as an example to be aspired to and a resource to be drawn on by other PHIs when advice or real-life examples of how to carry out HIAs in practice are needed [19,20,24,42]. WHIASU is currently globally unique in that it is the only dedicated HIA support unit based in a PHI in the world. It also highlights the need for a sustained, consistent approach to HIA methods and tools which Wales promotes at all levels of government and public health.

The findings reinforce that national and regional PHIs have a key role to play in both promoting and facilitating the use of HIAs within their contexts, either through setting a direction of travel by providing HIA guidance and tools or providing actual practical advice, guidance and some support, such as the model in Wales. In Wales, the HIA unit supports practitioners by mentoring and ‘learning by doing’ in partnership with public bodies and third-sector organisations and providing guidance, training and tools [24]. The Welsh model also promotes the idea that HIAs are not a ‘hard to do’ technical process in which one person must carry out all aspects of the HIA, but rather promotes the creation of multi-skilled, multidisciplinary teams to carry them out within an agency or organisation [43]. This approach demolishes the misconception that one person should conduct an HIA and promotes a time- and resource-effective and efficient HIA model of working—not dissimilar to how environmental assessment teams are constructed. This division also adds to the process, encourages a diverse range of perspectives and can lead to stronger working relationships. It can also support, through a division of labour approach, those who want to promote HIAs or review or carry out assessments but do not feel they have the confidence, time or financial resources.

HIAs of COVID-19 related response measures, for example, lockdowns [19,44], were also cited by several respondents and interviewees, and they believed these demonstrated the value of HIAs as a tool to be utilised by PHIs to capture the wider health impacts of plans and policies on equity in an evidence-based way. The increased awareness of the HIA as a process to better understand the wider health and equity impacts clearly derived from these COVID-19 HIAs, for example, economic or social impacts, and also highlighted how policy decisions can directly impact population health [19], providing a platform on which to build and continue to promote and demonstrate the usefulness of HIAs and HiAP approaches to policy and decision makers when seeking evidence to inform actions including mitigating negative impacts and maximizing positive ones. The role of NPHIs in leading these as a mandated lead for public health is also viewed as important, as is their role in building capacity [19]. The HiAP approach of engaging with other sectors and stakeholders, avoiding harm to health (both directly and indirectly) and reducing inequalities during the pandemic recovery is more important than ever [45] and provides a window of opportunity for HIAs.

The results from this study identify the need for further work in this arena to improve the practical application of HIAs and methods in PHIs. It could include the development of bespoke HIA awareness-raising, training, case study examples and targeted briefing papers for PHIs. This links to the ability to communicate about HIAs or ‘sell’ them to key stakeholders and politicians to ensure a better understanding of both the process and its benefits. Key themes identified were the development of dedicated support units or officers, case studies, training, templates and tools to help support institutes. Benefits of networks such as IANPHI [4] and the World Health Organization’s Regions for Health Network were not mentioned specifically but could also be explored, as they could be excellent vehicles for PHIs to improve their practice or knowledge, along with models such as the Welsh HIA Network of Practice [24], which could be replicated by other PHIs at a national or regional level.

There are limitations to this scoping research. The questionnaire was only disseminated in English and, therefore, limited the number of PHIs who participated and the geographical areas from which they came. This would, therefore, be reflected in the response rate. The sample for the survey were not representative of all international PHIs, and the majority of the survey responses were from the European region. However, a number of responses were submitted by participants from other regions, for example, Oceania. The response rate was 15.4% from 110 PHIs. Considering that the questionnaire was disseminated during the global COVID-19 pandemic and PHIs were, and still are in many cases, leading and focusing on the response to it, this is a respectable response rate. It must be noted that the researchers could not perform a cross-continent comparison due to the small sample size or carry out a comparison of country results. The analytical synthesis of the expert interview and survey results did, however, assist in confirming and boosting the findings and providing extra insight. The team were unable to outline the demographics of the interviewees, for example, age, due to the small sample size and potentially disclosing personal identifiable data on the participants.

Further research could support this study. It could include engaging with additional PHIs to boost the findings, uncover more insights, enable regional analysis and identify any differences or commonalities. Whilst Europe was strongly represented in the scoping survey respondents, the lack of responses from others such as North America and Africa could be explored to unpick this further. This could be due to regulatory context [28,30,31] or a focus on other public health challenges such as communicable diseases [4]. This exploratory work could also be used as a platform to better understand how public health institutes can utilize the support of different actors and agencies to raise awareness of, or enable the use of, HIAs. This could include carrying out additional scoping reviews with, for example, schools of public health in academic institutions or the WHO. It could also deepen investigations into the practice and use of HIAs in PHIs within their own contexts or regions.

There is a need for PHIs and HIA practitioners to support each other and share experiences, and HIAs could be a vehicle to facilitate this along with health protection expertise and health intelligence. Two impacts of this work to date have been better networking, including workshops at international conferences [46], and the replication of the survey, which has been translated and adapted at a local level in Portugal by the Portuguese Public Health Institute [47] to better understand and inform practice there.

## 5. Conclusions

This paper outlines the results from an online survey and interviews undertaken with representatives from PHIs to inform future practice in HIAs for PHIs and to share learning from each other. It develops a base for a shared understanding, paints an international picture of HIA practice, and can lead to future work at a global and national level. It adds to the evidence base around the practice and understanding of HIAs as a concept to mobilise HiAP. Additionally, it adds value by directly engaging with international PHIs—something not explored until now. PHIs have an important role in addressing the needs of their stakeholders, including public bodies such as local government and public health departments, by providing strategic direction for HIAs along with national governmental departments such as those in the health and environment fields. They can also provide HIA tools and resources such as those produced by Wales, Ireland and Scotland. The results from this study can also serve as a platform to help build knowledge, networks and expertise to promote capturing the co-benefits of investing in HIAs. This has the potential to encourage decision- and policy-makers to see health and well-being not as a cost, but as an investment that is the foundation of productive, resilient and stable economies, linking in with the HiAP approach and supporting them to address inequalities and ‘wicked issues’ which exist in all societies [12,48].

## Figures and Tables

**Table 1 ijerph-19-13367-t001:** Survey responses—barriers and benefits of HIAs.

Perceived Barriers	Perceived Benefits
Lack of resources (*n* = 12) Lack of capacity (*n* = 10) Not a priority at present for the institute or government (*n* = 7) Lack of training (*n* = 6) Lack of knowledge (*n* = 4) Lack of ability to advocate for HIAs (*n* = 2)	Focuses on prevention as an ‘ex-ante’ tool (*n* = 6) Can facilitate HiAP by considering health in other sectors (*n* = 5) Addresses equity/inequalities by considering population groups (*n* = 5) HIAs can improve plans, strategies and projects to make them healthier (*n* = 5) HIAs can have a focus on wider determinants of health (*n* = 4) Builds health into decision making (*n* = 2) HIAs can be a useful involvement tool through which to involve stakeholders (*n* = 2)

## Data Availability

The data presented in this study are available on request from the corresponding author. The data are not publicly available due to privacy reasons.

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
