# Peer review of "Facilitators, Barriers and Views on the Role of Public Health Institutes in Promoting and Using Health Impact Assessment—An International Virtual Scoping Survey and Expert Interviews"

_ijerph, 2022, doi:10.3390/ijerph192013367_

Round 1
Reviewer 1 Report
This is a well-written manuscript addressing the always relevant issue of the use and applicability of HIA and its benefits. It uses a well-developed and applied methodology. There are however several aspects that need to be revised and addressed before considering publication. Please see comments for specific sections below:
Abstract
Line 14 -it might need better support to assume PHI embrace the need to advocate for HIAP
Line 16 -define acronyms when mentioned the first time.
Line 18 – nine representatives of who/what?
Line 19 – what if the references of this figures? (11=64.7% but you mentioned 9 representatives? – the method is not clear)
Line 20 – you refer to “Assessments” -which (other) ones?
Lines 23 –“act” is not the best word here
Introduction.
Please see some specific comments:
Lines 48-50. Citation needed
Lines 54-56. This is assigning a new dimension of duties to PHI (“advocating for health and well-being”). Need to be supported (relevant literature).
line 62 -needs more support (Citation 6 is a report but there is not clear how widely accepted/implemented the concept is embraced by PHI (from the text: “it demonstrates that HiAP is feasible in different contexts and provides fresh insight into how to seize opportunities to promote HiAP and how to implement policies for health across sectors”).
Lines 89-93 -might be better place in the conclusions
Methods
The methods are clearly provided and explained. The only comment is on the need of ethics approvals especially if there are recorded interviews. Even if the NHS -RA tool does not suggest the need of ethics approvals, this is an important consideration. I suggest this section includes further explanation on the conduction of the research without ethics approval and mechanisms to protect any low-risk of discomfort to participants
Results
Gral comment: Since the results present the outputs by survey question, you can consider a table with the these, including percentages while keeping the narrative corresponding to the interviews outcomes. This would help to visualise a big picture of the results as well as specific details for each question.
Some specific comments:
Line 203. -you probably meant “interviewees from 4 countries”. Also, not sure what is meant by “environmental field of application”?
Line 206 -please add which countries were these
Discussion
Some specific comments:
Line 368. -since there were not answers from America (not just the US) I wouldn’t assume this as global
Lines 375-378. -from the results this is not exactly clear (or the results need to be re-focused to highlight this)
Line 378. –“ This is consistent with previous surveys” reads as an orphan sentence -need to expand further on this
Line 379. -it seems the discussion focuses on differences between countries, but this is not consistent with the approach presented in the results (maybe a table with specific differences per country can be included). Consider a comment above on the use of tables in the results -as it currently stands, the results read more like a summary of outcomes per question rather than an analysis comparing these against some other aspects (such as differences between countries, socioeconomic contexts or different industries linked).
Lines 295-402. This addresses lightly an aspect that needs to be expanded in the discussion, especially considering HIA is just part of EIAs in many countries. Should consider some missing relevant literature: doi 10.1186/1472-6963-14-371; doi: 10.1515/reveh-2019-0033;
Line 406. Citation needed. Some important literature on this is missing, for example: doi: 10.1136/jech.2004.026039; \\
Line 420. This needs to consider that it would require more than just promotion and collaborative work within PHI. Suggest discussing on the reasons why HIA has not been integrated as a supportive tool compared to other impact assessment methods (such as EIA or even SIA).
Parags. Lines 436-462. It’s hard not to feel the discussion narrows on one region in a single European country that strongly embraces HIA. While this highlights the strengths of HIA in some areas in Europe especially, it also calls attention on the little impact it has on other regions, especially the US. This section can expand on these issues and the reasons for this, to explore how HIA applicability can be increased globally.
Line 464 –“measures”
Parag. 464-471. This parag. can discuss/incorporate some important highlights of a paper cited in the manuscript (citation 15).
Author Response
Reviewer 1: |
|
Abstract Line 14 -it might need better support to assume PHI embrace the need to advocate for HIAP Line 16 -define acronyms when mentioned the first time. Line 18 – nine representatives of who/what? Line 19 – what if the references of this figures? (11=64.7% but you mentioned 9 representatives? – the method is not clear) Line 20 – you refer to “Assessments” -which (other) ones? Lines 23 –“act” is not the best word here |
Thank you – the abstract has been amended to reflect the comments made. |
Introduction. Please see some specific comments: Lines 48-50. Citation needed Lines 54-56. This is assigning a new dimension of duties to PHI (“advocating for health and well-being”). Need to be supported (relevant literature). line 62 -needs more support (Citation 6 is a report but there is not clear how widely accepted/implemented the concept is embraced by PHI (from the text: “it demonstrates that HiAP is feasible in different contexts and provides fresh insight into how to seize opportunities to promote HiAP and how to implement policies for health across sectors”). Lines 89-93 -might be better place in the conclusions |
Thank you.
These have now been amended and citations added.
The content in lines 89-93 has also been summarised in the conclusion to strengthen that. |
Methods The methods are clearly provided and explained. The only comment is on the need of ethics approvals especially if there are recorded interviews. Even if the NHS -RA tool does not suggest the need of ethics approvals, this is an important consideration. I suggest this section includes further explanation on the conduction of the research without ethics approval and mechanisms to protect any low-risk of discomfort to participants |
Thank you. We do not feel like we need to add any further detail to this section as we have said clearly that the study posed little potential harm to those taking part and all the data which was collected and analysed was anonymised and safely secured digitally to protect personal data and privacy. We have also stated that informed consent was collected for interview participation, and the nature of the research was not sensitive. The only thing we have added is that the interviews were anonymised on the point of transcription. |
Results Gral comment: Since the results present the outputs by survey question, you can consider a table with the these, including percentages while keeping the narrative corresponding to the interviews outcomes. This would help to visualise a big picture of the results as well as specific details for each question. Some specific comments: Line 203. -you probably meant “interviewees from 4 countries”. Also, not sure what is meant by “environmental field of application”? Line 206 -please add which countries were these
|
Thank you.
We are content to keep the results section as it is currently formatted with both the survey and interview results intertwined, as they complemented each other throughout the analysis.
Line 203 has been amended to add clarity.
Including the information in line 206 could compromise the anonymity of the interviewees and therefore we have not included this level of detail. |
Discussion Some specific comments: Line 368. -since there were not answers from America (not just the US) I wouldn’t assume this as global Lines 375-378. -from the results this is not exactly clear (or the results need to be re-focused to highlight this) Line 378. –“ This is consistent with previous surveys” reads as an orphan sentence -need to expand further on this Line 379. -it seems the discussion focuses on differences between countries, but this is not consistent with the approach presented in the results (maybe a table with specific differences per country can be included). Consider a comment above on the use of tables in the results -as it currently stands, the results read more like a summary of outcomes per question rather than an analysis comparing these against some other aspects (such as differences between countries, socioeconomic contexts or different industries linked). Lines 295-402. This addresses lightly an aspect that needs to be expanded in the discussion, especially considering HIA is just part of EIAs in many countries. Should consider some missing relevant literature: doi 10.1186/1472-6963-14-371; doi: 10.1515/reveh-2019-0033; Line 406. Citation needed. Some important literature on this is missing, for example: doi: 10.1136/jech.2004.026039; \\ Line 420. This needs to consider that it would require more than just promotion and collaborative work within PHI. Suggest discussing on the reasons why HIA has not been integrated as a supportive tool compared to other impact assessment methods (such as EIA or even SIA). Parags. Lines 436-462. It’s hard not to feel the discussion narrows on one region in a single European country that strongly embraces HIA. While this highlights the strengths of HIA in some areas in Europe especially, it also calls attention on the little impact it has on other regions, especially the US. This section can expand on these issues and the reasons for this, to explore how HIA applicability can be increased globally. Line 464 –“measures” Parag. 464-471. This parag. can discuss/incorporate some important highlights of a paper cited in the manuscript (citation 15). |
Line 368 has been amended.
Lines 375-378 have been amended to reflect the comment.
Line 378 has been amended.
Line 379. – the paragraph has been amended to reflect the comment. Unfortunately due to the small sample size, we are unable to present a cross-country comparison as would potentially allow for identification of individuals.
Lines 295-402 this has been amended to reflect these comments and also later in the discussion about how to move forward and overcome some barriers.
Line 406 the citation has been added – thank you.
Line 420 this has been expanded on.
Lines 436 – 462 this has been addressed in the discussion.
464-471 this paragraph has been expanded to incorporate this.
|
Reviewer 2 Report
This is timely and important survey of gaps in Health Impact Assessment with an international focus. I recommend it for publication and see it to be a significant contribution to the literature.
In terms of areas where there could be potential improvement, while the article defines Health Impact Assessment, I was unclear as to the types and scope of actual projects that were being done in the regional cases (Wales, Scotland, Portugal etc.). In it's current form the piece is detailed in that there are interview quotes and the methodology is clear. However, empirical evidence could be improved with more project details.
A thought, in terms of an area for improvement, would be to provide more details in terms of the types of reviews that are being done to ground the piece more concretely. This would make the article less high level and more relevant. Ie. what impacts are being analyzed in the HIAs and what health outcome gaps are not being met due to the lack of resources for HIAs?
Further, another site of future research might include the North American context. Again, this is overall a significant contribution and I appreciate the addition to the field of HIA research.
Author Response
Reviewer 2: |
|
This is timely and important survey of gaps in Health Impact Assessment with an international focus. I recommend it for publication and see it to be a significant contribution to the literature In terms of areas where there could be potential improvement, while the article defines Health Impact Assessment, I was unclear as to the types and scope of actual projects that were being done in the regional cases (Wales, Scotland, Portugal etc.). In it's current form the piece is detailed in that there are interview quotes and the methodology is clear. However, empirical evidence could be improved with more project details. A thought, in terms of an area for improvement, would be to provide more details in terms of the types of reviews that are being done to ground the piece more concretely. This would make the article less high level and more relevant. Ie. what impacts are being analyzed in the HIAs and what health outcome gaps are not being met due to the lack of resources for HIAs? |
Thank you. Specific projects weren’t specified within the interviews of this study, but would be a great point for future research which could do further deep dives into current practices. A note has been added into the discussion on this. |
Further, another site of future research might include the North American context. Again, this is overall a significant contribution and I appreciate the addition to the field of HIA research. |
Thank you.
This has been added into the future research section of the discussion. |
Reviewer 3 Report
The title does match much of the paper. The title as it currently stands is “The role of Public Health Institutes in promoting and using Health Impact Assessment”. Much of the paper focuses on the barriers and benefits of PHIs using HIA in their daily activities. This is inconsistent with the title. Further, the introduction and conclusion of the paper talk about the use of HIA in a HiAP approach. The rest of the paper is silent on this.
The Discussion section could be strengthened by focusing on the barriers with advice and recommendations on how to reduce these so that PHIs can more frequently employee a HIA. Currently, the Discussion section expands on the items gleaned from the interviews, but a more pragmatic approach that allows the reader to learn how to be more involved in HIAs could be helpful. Reducing the barriers by promoting and benefits should be the central theme of the Discussion.
Author Response
Reviewer 3: |
|
The title does match much of the paper. The title as it currently stands is “The role of Public Health Institutes in promoting and using Health Impact Assessment”. Much of the paper focuses on the barriers and benefits of PHIs using HIA in their daily activities. This is inconsistent with the title.
|
Thank you. The title has been amended to reflect this comment. |
Further, the introduction and conclusion of the paper talk about the use of HIA in a HiAP approach. The rest of the paper is silent on this. |
HIAP is included in the Results and Discussion sections, but this has been strengthened and hopefully made much clearer. |
The Discussion section could be strengthened by focusing on the barriers with advice and recommendations on how to reduce these so that PHIs can more frequently employee a HIA. Currently, the Discussion section expands on the items gleaned from the interviews, but a more pragmatic approach that allows the reader to learn how to be more involved in HIAs could be helpful. Reducing the barriers by promoting and benefits should be the central theme of the Discussion.
|
This is a good point and some sentences have been added into the Discussion section to reflect these comments. |
Reviewer 4 Report
The authors present an original investigation throughout the use of a questionnaire and semi-structured interviews.
Due to the limited answers, the study findings are necessarly weak; therefore, unfortunately, the results are not "representative".
Nonetheless, the results are interesting, and this type of research is much needed in public health, mostly to improve what PHI can offer to the general population.
I suggest acceptance of the present manuscript, after two minor adjustments.
1. I find the 45-item questionnaire to be of importance, and it should be presented by the authors. If it has not been uploaded (maybe as supplementary material), it should be, otherwise the research could not be considered reproducible, and therefore not acceptable by modern science standards.
2. The interviewed participants should be disclosed at the very least by presenting a table which should illustrate:
Country of origin, gender, age.
It would be nice to have also listed the affiliation of said partecipants, without disclosing any personal information of course.
This table is necessary because different people bring different perspectives, and it is possible aged collegues might feel differently than more young collegues.
After these two suggestions will be implemented, the manuscript could be accepted for publication.
In thanking the authors for their work, I wish everyone involved in this process a great week.
Author Response
Reviewer 4: |
|
The authors present an original investigation throughout the use of a questionnaire and semi-structured interviews.
Due to the limited answers, the study findings are necessarly weak; therefore, unfortunately, the results are not "representative".
Nonetheless, the results are interesting, and this type of research is much needed in public health, mostly to improve what PHI can offer to the general population.
I suggest acceptance of the present manuscript, after two minor adjustments.
|
Thank you for your comments. A sentence has been added to the limitations in the discussion sentence which recognises that the survey was not representative, purely a scoping survey. |
I find the 45-item questionnaire to be of importance, and it should be presented by the authors. If it has not been uploaded (maybe as supplementary material), it should be, otherwise the research could not be considered reproducible, and therefore not acceptable by modern science standards.
|
The questionnaire has been added as Supplementary Material. |
The interviewed participants should be disclosed at the very least by presenting a table which should illustrate: Country of origin, gender, age.It would be nice to have also listed the affiliation of said partecipants, without disclosing any personal information of course. This table is necessary because different people bring different perspectives, and it is possible aged collegues might feel differently than more young collegues. After these two suggestions will be implemented, the manuscript could be accepted for publication. In thanking the authors for their work, I wish everyone involved in this process a great week.
|
The research team discussed this prior to submitting the manuscript and believe in this scenario it is not appropriate to disclose further demographic information on the interview participants. This is due to the following reasons; age and gender were not captured through the interview process; disclosure of affiliation will potentially allow for participants to become identifiable due to the niche field and the small number of HIA practitioners who are currently situated in Public Health Institutes. Lines 204-207 try to provide some information on the participants but without identifying them.
We have also added a sentence to the discussion to reflect that a limitation of the work was not being able to analyse or disclose participant demographics further than what we have due to caveats around disclosing identifiable information. |
Round 2
Reviewer 3 Report
Amendments were appropriate.